



# Spatiotemporal clustering of flash floods in a changing climate (China, 1950-2015)

Nan Wang[1,2], Luigi Lombardo[3], Marj Tonini[4], Weiming Cheng[1,2,5,6,*], Liang Guo[7,8], Junnan Xiong[1,9]

## Abstract

The persistence over space and time of flash flood disasters – flash floods that have
caused either economical or life losses, or both – is a diagnostic measure of areas subjected
to hydrological risk. The concept of persistence can be assessed via clustering analyses,
performed here to analyse the national inventory of flash flood disasters in China occurred in
the period 1950-2015. Specifically, we investigated the spatiotemporal pattern distribution
of the flash flood disasters and their clustering behavior by using both global and local
methods: the first, based on the Ripley's K-function, and the second on Scan Statistics. As
a result, we could visualize patterns of aggregated events, estimate the cluster duration and
make assumptions about their evolution over time, also with respect precipitation trend.
Due to the large spatial (the whole Chinese territory) and temporal (66 years) scale of the
dataset, we were able to capture whether certain clusters gather in specific locations and
times, but also whether their magnitude tends to increase or decrease. Overall, the eastern
regions in China are much more subjected to flash flood disasters compared to the rest of
the country. Detected clusters revealed that these phenomena predominantly occur between
July and October, a period coinciding with the wet season in China. The number of detected
clusters increases with time, but the associated duration drastically decreases in the recent
period. This may indicate a change towards triggering mechanisms which are typical of
short-duration extreme rainfall events. Finally, being flash flood disasters directly linked to
precipitation and their extreme realization, we indirectly assessed whether the magnitude
of the trigger itself has also varied through space and time, enabling considerations in the
context of climatic changes.

[1]State Key Laboratory of Resources and Environmental Information Systems, Institute of Geographic Sciences and Natural Resources Research, Chinese Academy of Sciences, Beijing, 100101, China

[2]University of Chinese Academy of Sciences, Beijing, 100049, China

[3]University of Twente, Faculty of Geo-Information Science and Earth Observation (ITC), PO Box 217, Enschede, AE 7500, Netherlands

[4]Institute of Earth Surface Dynamics, Faculty of Geosciences and Environment, University of Lausanne, CH-1015 Lausanne, Switzerland

[5]Jiangsu Center for Collaborative Innovation in Geographic Information Resource Development and Application, Nanjing, 210023, China

[6]Collaborative Innovation Center of South China Sea Studies, Nanjing, 210093, China

[7]Research Center on Flood and Drought Disaster Reduction of the MWR, Beijing, 100038, China

[8]State Key Laboratory of Simulation and Regulation of Water Cycle in River Basin, China Institute of Water Resources and Hydropower Research, Beijing 100038, China

[9]School of Civil Engineering and Architecture, Southwest Petroleum University, Chengdu, 610500, China





**Keywords:** Spatiotemporal clustering; Flash flood disasters; Precipitations; China

# 1    Introduction

Flash floods are among the most destructive surface processes around the world, especially in mountainous areas (Au, 1998; Borga *et al.*, 2011; Gomez and Kavzoglu, 2005; Jonkman, 2005). They are mainly initiated by rapid and intense rainfall, often discharged in few hours (e.g., Borga *et al.*, 2007; Bout *et al.*, 2018; He *et al.*, 2018; Lóczy *et al.*, 2012), and by complex interactions of the climatic conditions with topography and hydrology (e.g., Hatheway *et al.*, 2005). Because of the very rapid raise in water levels caused by flash floods, it is challenging to take timely and effective actions to contain the associated damage. Flash flood disasters are essentially flash floods that have caused losses either in terms of human lives or economy, or both (Gaume *et al.*, 2009; Jonkman and Kelman, 2005; Kelman and Spence, 2004). In China, approximately 70% of the total area is covered by mountains and hills, which exposes a substantial surface of the national territory to flash flood disasters' risk (Liu *et al.*, 2018). Additionally, the more frequent extreme precipitation associated with climate change has increased the number of flash flood disasters in recent decades (Sampson *et al.*, 2015).

Historical inventories of flash flood disasters are a precious source of information allowing to investigate their spatiotemporal pattern distribution and evolution. Furthermore, this information can be related with the geomorphological setting of the area and the climatic/meteorological conditions to detect triggering factors, highlight the more vulnerable areas, and to prevent and forecast their effects in the future.

The susceptibility to hydro-geomorphological processes is commonly assessed by considering only the spatial distribution of observed events (Cama *et al.*, 2015, 2017; Santangelo *et al.*, 2012; Zaharia *et al.*, 2017). However, this is purely a convenient assumption from the modeling perspective. Recently, a growing amount of evidence indicates that these events tend to aggregate in space conditioned by the temporal variability, attesting for an interaction between space and time on event frequency and distribution (Gariano and Guzzetti, 2016; Kouli *et al.*, 2010; Zhang and Cong, 2014; Fuchs *et al.*, 2015; Merz *et al.*, 2016; Tonini and Cama, 2019). In other words, when an event occurs at a specific location, a temporary increase in the probability that other events will cluster at nearby locations should be accounted for. This increase in probability can be captured through clustering analyses and various examples already exist in literature where this has been done at different spatial and temporal scales and via different analytical approaches. Notably, this type of application spans in many areas of natural hazards and have become mainstream in case of seismicity (e.g., Fischer and Horálek, 2003; Georgoulas *et al.*, 2013; Varga *et al.*, 2012; Woodward *et al.*, 2018; Yang *et al.*, 2019), joint sets and their orientation in rock outcrops (e.g., Tokhmechi *et al.*, 2011; Zhan *et al.*, 2017), groundwater monitoring (Chambers *et al.*, 2015), wildfires (e.g., Orozco *et al.*, 2012; Costafreda-Aumedes *et al.*, 2016; Fuentes-Santos


*et al.*, 2013; Tonini *et al.*, 2017), and landslides (e.g., Lombardo *et al.*, 2018, 2019a; Tonini
and Cama, 2019). In the specific case of flooding, Zhao *et al.* (2014) used the projection
pursuit theory to cluster spatial data and to build a dynamic risk assessment model for flood
disasters. Moreover, Renard (2017) detected flood vulnerability accounting for clustering
effects in key areas with high flood risk. Pappadà *et al.* (2018) also investigated the flood
risks in a given region and identified clusters where the floods show a similar behavior with
respect to multivariate criteria. Gu *et al.* (2016a,b) indicated the floods in Tarim River basin
showed evident inter-annual clustering pattern. Another example can be found in Merz *et al.*
(2016) where the authors analyzed the inter-annual and intra-annual flood clustering in Ger-
many. All these examples confirm a substantial scientific interest in recent years dedicated
to investigate the clustering behaviors of flash floods and the associated risk; and, more
generally, to concurrently analyze their spatial and temporal persistence. However, despite
the scientific efforts, detecting flash flood patterns at long temporal scale is still scarce in
literature, mainly because of technical limitations. In fact, limited information and records
are available in digital form reporting locations and dates of flash floods (and flash flood
disasters), especially over long periods. Nevertheless, very recent advances in data collection
and sharing techniques are gradually filling this gap, and an increasing number of databases
are being published and made available to the scientific community with the records of his-
torical and hydro-geomorphological disasters at the global, continental, or regional scale over
long periods (Archer *et al.*, 2019; de Bruijn *et al.*, 2019; Gourley *et al.*, 2013; Haigh *et al.*,
2017; Nowicki Jessee *et al.*, 2020; Vennari *et al.*, 2016; Wood *et al.*, 2020).
Typically, flash flood disasters (as many other hydro-geomorphological disasters) can be
considered as a stochastic point processes (Stoyan, 2006) acting in both spatial and tempo-
ral dimensions (e.g., Lombardo *et al.*, 2019b). Point patterns can be analyzed in terms of
their random distribution, dispersion and clustering behaviour (Merz *et al.*, 2016; Tonini and
Cama, 2019). Several methods can be implemented to deal with stochastic properties. Some
classic models, such as Moran's I (Moran, 1950), Ripley's K-function (Ripley, 1977), fractal
dimension (Lovejoy *et al.*, 1986), and Allan factor (Allan, 1966), have been used to detect
clustering behaviour in space and in time. Representative models for local clustering analysis
(i.e. allowing to detect clusters and their specific location) include Geographical Analysis
Machine (GAM, Openshaw *et al.*, 1987), Turnbull's Cluster Evaluation Permutation Proce-
dure (CEPP, Turnbull *et al.*, 1990), Scan Statistics (Kulldorff, 1997), and DBSCAN (Ester
*et al.*, 1996). For flash floods, which are triggered by storms, the temporal dependency among
persistent events is mainly driven by climatic and meteorological conditions. However, global
cluster indicators only take into consideration one dimension, disregarding the interaction
between space and time. In this sense, spatiotemporal Scan Statistics is a good tool to detect
clusters since it allows to identify statistically significant excess of observations thanks to a
moving cylindrical window that scans all locations both in space and time (Kulldorff *et al.*,
1998). Therefore, it is especially useful to investigate hydro-geomorphological processes such
as flash floods. For such phenomena, the detection of events aggregated over a given region




and in a specific period, generally yields more informative results than the purely spatial or temporal analysis. Furthermore, understanding the magnitude of the persistence for flash flood disasters is an important requirement to predict where, when and how their highest probability to occur distributes in the future.

In this study, we explored the spatiotemporal pattern distribution of flash flood disasters which have caused either or both life and economic losses in China over the period 1950-2015. Firstly, the deviation of flash flood disasters from a spatiotemporal random process is explored by applying the spatiotemporal Ripley's K-function. Then, the Scan Statistics was applied to detect statistically significant spatiotemporal clusters. Finally, the possible relationship between the detected clusters and local climatic proxy factors is discussed. To the best of our knowledge, it is the first time that such a long-term inventory is analysed to explore the spatiotemporal patterns of flash flood disasters, especially in China. This study provide useful insights on flood dynamics over a large spatiotemporal domain. Moreover, because of the long time-span, results can be useful to indicate how flash flood disasters have evolved in response to climate changes.

# 2    Material and methods

## 2.1    Data description

### 2.1.1    Study area

China lies between latitudes 18° and 54° N, and longitudes 73° and 135° E. With an area of about 9.6 million square kilometers, it is the world's third-largest country. The landscape varies significantly across this vast area, ranging from the Gobi and Taklamakan deserts in the north to the subtropical forests in the wetter south. The eastern plains and southern coasts are the location of most of China's agricultural land and settlements. The southern areas consist of hilly and mountainous terrain. The west and north of the country are dominated by sunken basins (such as the Gobi and the Taklamakan desert), towering massifs and rolling plateaus, including part of the highest tableland on earth, the Tibetan Plateau. Based on its topography, China can be divided into six homogeneous geomorphological macro-regions (Wang *et al.*, 2020): eastern plain, southeastern hills, southwestern mountains, north-central plains, northwestern basins and Tibetan Plateau. Mountains (33% of the territory), plateaus (26%) and hills (10%) account together for nearly 70% of the entire surface.

In recent years, the precipitation intensity shows an increasing trends over China (Zhang and Cong, 2014). Influenced by the East Asian summer monsoon and the geomorphologic settings, the climatic condition across the whole country varies considerably (Wu *et al.*, 2019). In general, the wet season in China lasts from May to September (Song *et al.*, 2011b). In the Eastern area, the annual rainfall decreases from south to north with an average annual precipitation that ranges from 250 to 750 $mm$ (Zhang *et al.*, 2007). In the west and central part of North China, due to its far distance away from ocean, the climate tends to be more
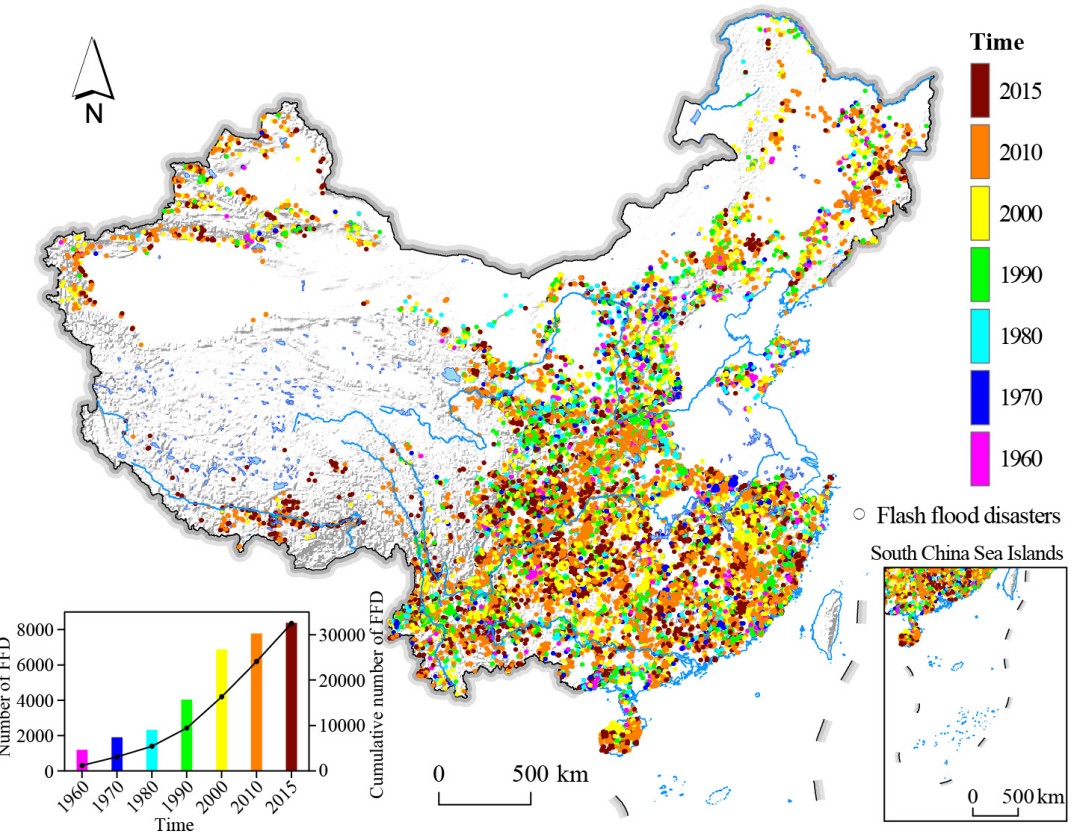

Figure 1: Distribution of flash flood disasters and background setting of China.

arid and the landscape transitions to large deserts. The Tibetan plateau is characterized by wet and humid summers with cool and dry winters. More than 60–90% of the annual total precipitation falls between June and September (Xu *et al.*, 2008).

### 2.1.2 Flash flood disasters inventory

The dataset used in this study has been collated and made accessible for the present research as part of a national effort carried out by the Chinese Institute of Water Resources and Hydropower Research (Liu *et al.*, 2018). It reports flash flood occurrences in China since 1950 until 2015 together with available information, namely longitude and latitude, date, fatalities and economic losses. Due to the lack of specific terminology and/or detailed descriptions of the disaster process in the database, the data does not differentiate the initial mechanism, be it water floods or debris floods/flows (e.g., Fernández and Lutz, 2010; Gartner *et al.*, 2014). The only common information is that for each specific case, a large amount of overland flows, mixed with an unspecified solid fraction, rapidly flooded a given area with disastrous effects




(e.g., Chang *et al.*, 2011; Pierson *et al.*, 1987).
To better understand the spatiotemporal dynamics of flash floods and associated disas-
ters, as well as the relationship with the triggering factors, the date of occurrence is of vital
importance. Therefore, for consistency reasons, we considered only the records whose meta-
data contained a full temporal description (year-month-day) resulting in a subset of 32,473
flash flood disasters (accounting for 68% of the entire dataset) precisely located in space and
time (Figure 1). We further defined the impact of flash flood disasters as the combination
of fatalities and economic losses (see Table 1), and we refer to this classification throughout
the manuscript.

Table 1: Impact of flash flood disasters (RMB = renminbi, the official currency of China).

| Economic Loss | Number of Fatalities | | | | | |
|---|---|---|---|---|---|---|
| ($10^4$RMB) | 0 | 0-5 | 5-10 | 10-50 | 50-100 | $\geqslant$100 |
| 0 | -- | F1 | F2 | F3 | F4 | F5 |
| 0-100 | F1 | F1 | F2 | F3 | F4 | F5 |
| 100-1000 | F2 | F2 | F2 | F3 | F4 | F5 |
| 1000-10000 | F3 | F3 | F3 | F3 | F4 | F5 |
| 10000-100000 | F4 | F4 | F4 | F4 | F4 | F5 |
| $\geqslant$100000 | F5 | F5 | F5 | F5 | F5 | F5 |

## 160 2.2    Methodological overview

### 161 2.2.1    Spatiotemporal K-function

The Ripley's K-function ($K_{(s)}$) is largely applied in environmental studies to analyse the
pattern distribution of spatial point processes and to detect deviation from spatial random-
ness. $K_{(s)}$ allows to determine if a set of mapped punctual events show a random, dispersed
or cluster distribution pattern over increasing distance values (Ripley, 1977). It is computed
as the ratio between the expected number of events falling at a distance $r$ from an arbitrary
event and the average number of points per unit area, corresponding to the intensity of the
spatial point process ($\lambda$). In the same way, it is possible to define the temporal K-function
($K_{(t)}$) allowing to asses for the randomness of events in time. The spatiotemporal K-function
($K_{(s,t)}$) is a generalization of the univariate Repley's K-function which allows to test for the
independence between two variables, space ($s$) and time ($t$). Therefore, the $K_{(s,t)}$ is a suitable
tool to investigate the clustering behaviour of a set of events occurred in a given area at a
given time. For a point process $X$ with intensity $\lambda$, according to equation 1, it is defined as
the number of expected further events ($E$) occurring within a distance $r$ and time $t$ from an
arbitrary event $u$, where $a$ define the contouring circle.


$$K_{(s,t)} = 1/\lambda \times E[n(X \cap a(u,r,t)u)|u \in X] \tag{1}$$

To illustrate the interaction between space and time, it can be useful to evaluate the value $D_{(s,t)}$, defining the difference between the spatiotemporal K-function and the product of the purely spatial and the purely temporal K-function (see equation 2).

$$D_{(s,t)} = K_{(s,t)} - K_{(s)} \times K_{(t)} \tag{2}$$

If space and time are independent variables, this value equals to zero. Otherwise, positive values of $D_{(s,t)}$ indicates the interaction among events in space and in time. In other words, events closer in space are more likely to occur in a closer time. On the contrary, the negative values means a dispersed pattern.

In this study, spatiotemporal K-function analyses were performed with the package "Spatial and Space-Time Point Pattern Analysis" (splancs, Rowlingson and Diggle, 2017) in R (R Team _et al._, 2019).

### 2.2.2 Spatiotemporal scan statistics

Scan statistic was originally developed by Naus (1965a,b) to detect cluster in a one-dimensional point process. Subsequently Kulldorff (1997) extended this approach to multi-dimensional point process, introducing the use of scanning windows. The procedure was implemented into a free software, SaTScan[TM] (satscan.org) which can handle a purely spatial, purely temporal or spatiotemporal datasets and includes different probability models depending on the nature of the data and the scope of the research (e.g. for prospective or retrospective cluster detection). In the purely spatial case, the aim of scan statistics is the early detection of clusters, allowing one to map them and to assess their statistical significance. Moving windows scan the region increasing their radius up to a fixed limit ($R_{max}$) and count the number of events falling inside and outside the area. The probability that a window contains more observations than expected is assessed via the likelihood ratio, by comparing with the background population. Then, the null hypothesis of randomness is tested by Monte Carlo experiments, based on repeated random sampling. The spatiotemporal scan statistic use cylinders instead of circular windows, where the height of the cylinder account for the temporal dimension. In order to deal with flash foods, the retrospective spatiotemporal permutation scan statistics (STPSS, Kulldorff _et al._, 2005) seems to be the most adequate model. Indeed, for environmental processes, the definition of the background population at risk needed for the statistical significance assessment of the detected clusters is quite problematic. STPSS assesses the expected number of cases using only the observed cases by permutation, supposing that each event has the same probability for all the times. Computationally, if $C$ is the total number of observer cases and $c_{zd}$ the number of cases observed in a zone $z$ and a day $d$, the expected number of cases per zone and day ($\mu_{zd}$) is:




$$\mu_{zd} = \frac{1}{C} \left( \sum_z c_{zd} \right) \left( \sum_d c_{zd} \right) \tag{3}$$

It follows that, for a spatiotemporal cylinder $A$, the expected number of cases $\mu_A$ can be estimated as the sum of each $\mu_{zd}$ inside the cylinder $A$:

$$\mu_A = \sum_{z,d \in A} \mu_{zd} \tag{4}$$

If $C_A$ is the number of observed cases in $A$, considered as Poisson-distributed with mean $\mu_A$, the Poisson generalized likelihood ratio ($GLR$) can be computed as:

$$GLR = \left( \frac{c_A}{\mu_A} \right)^{c_A} \left( \frac{C - c_A}{C - \mu_A} \right)^{C - c_A} \tag{5}$$

This ratio is calculated and maximized for every possible scanning cylinder. Then, the Monte Carlo simulations are performed and the statistical significance of the detected clusters is assigned by ranking the clusters according to their $GLR$-value.

## 3    Results

### 3.1    Deviation from a random process

In the present study the spatiotemporal K-function is used to assess the interaction between the two variables, space and time, in generating clusters at increasing distances. Figure 2 (panel a) shows the 3D-plot of $D_{(s,t)}$ with a zoom up to 2000 $km$ (panel b). Positive values indicate that space and time interact in generating clusters: in other words, events closer in space are also closer in time. This is the case at any increasing distance, from hundred meters to thousands meters and from few years to decades.

In addition, we computed the spatiotemporal K-function separately for the eastern and western side of China (Figure 3). We did this because the southeastern area, which is the rainiest part of the country, is highly affected by flash floods, while the northwestern area is predominantly desert and flash floods are less frequent. It results that, although events are clustered in both the areas, in the southeastern area (panel a) clusters arise at a shorter spatial distance and closer in time than in the northwestern area (panel b). More specifically, in the southeast China the spatiotemporal interaction generates clusters starting from 200 $km$ and a plateau is reached at about 1800 $km$. In Northwest China the global cluster behaviour is more evident from about 1000 $km$ to higher distances. As regards the temporal dimension, the two part of the country show a similar cluster behaviour, with a strong attraction among events during the first 20 years lasting in time with a more relaxed clustering behaviour.

To summarize, the spatiotemporal K-function reveals a deviation of flash flood disasters and associated spatiotemporal pattern distribution from a random process at specific scales,

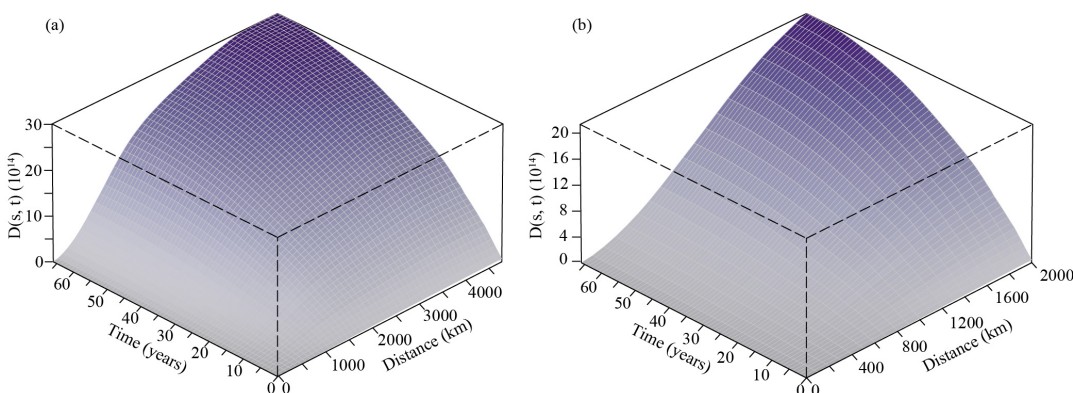

Figure 2: Three dimensional summary of flash flood disasters in China during 1950-2015.

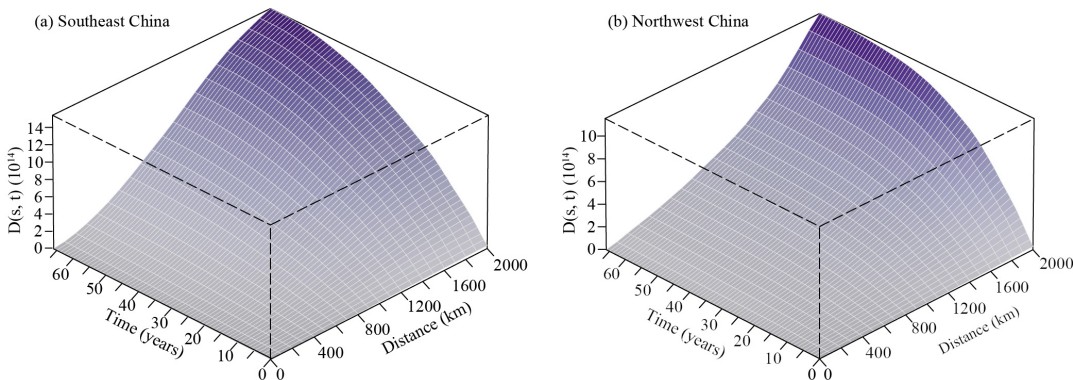

Figure 3: Three dimensional summary of flash flood disasters in China, separated between two eastern and western sectors and with a maximum spatial bandwidth of 2000 $km$.

measured and quantified both in space, as distances-values, and in time, as yearly periods.
These values can provide a useful indication to set up the parameters for further clustering
algorithms, acting at local scale such as, for example, the spatiotemporal scan statistics.

## 3.2 Spatiotemporal clusters

### 3.2.1 Cluster detection and spatial distribution

Scan statistics was performed to detect spatiotemporal clusters of flash flood disasters. The
size and the duration of the detected clusters are influenced by the input parameters of the
scanning windows, namely the maximum radius ($R_{max}$), the maximum temporal duration
($T_{max}$), and the time aggregation ($T_{agg}$). Indeed, values of $R_{max}$ exceeding the 50% of the
total area or, for $T_{max}$, the 50% of the entire study period, can result in an exceptionally low
rate outside the scanning window rather than detecting an exceptionally high rate inside.


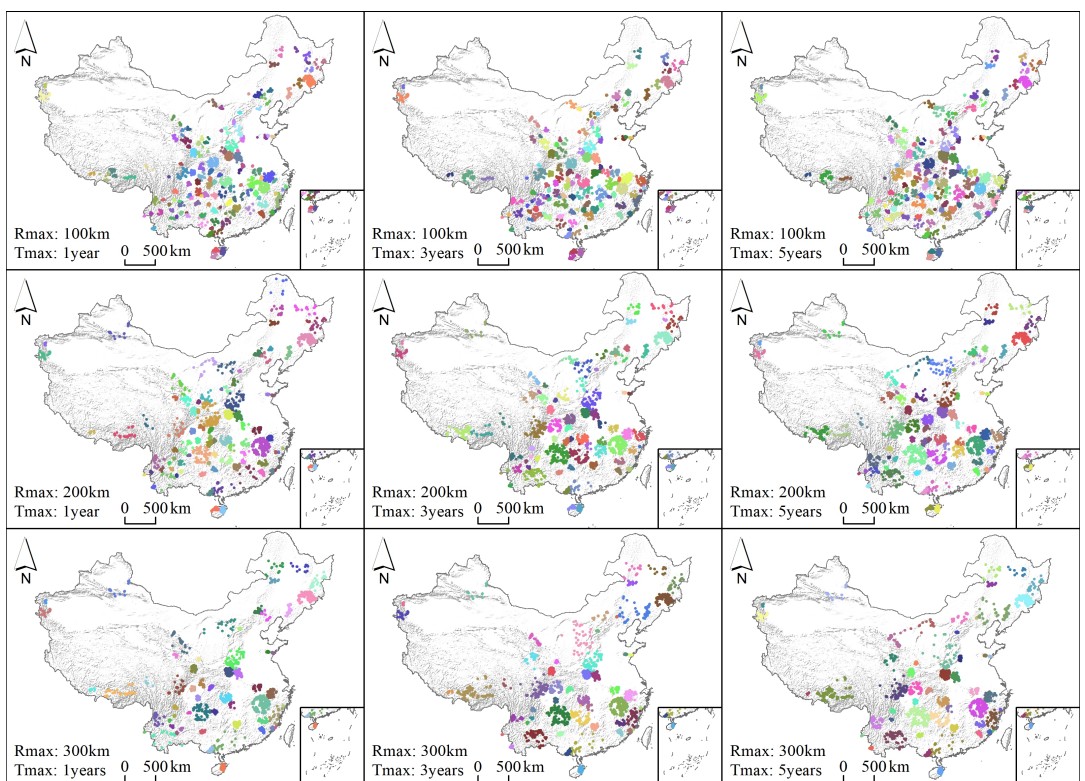

Figure 4: Significant (p<0.005) spatiotemporal clusters of flash flood disasters in China during 1950-2015.

$T_{agg}$ is used to adjust the aggregation of the data over time and allows adjusting for cyclic temporal trends: for example, a time aggregation of one year automatically adjusts for the seasonal variability, while the contrary happen with monthly aggregations. Moreover, both spatial and temporal aggregations can highly reduce the computer processing time. Following the results obtained by the spatiotemporal K-function and discussed above, few radii for each area (southeast and northwest China) were tested. Performed analyses indicated that the effect onto the detected clusters were negligible and finally we considered the spatiotemporal distribution of flash flood disasters as a whole rather than splitting the Chinese territory in two areas. We opted for a set of possible combinations of $R_{max}$ and $T_{max}$, keeping $T_{agg}$ fixed to one year. More specifically, to compare the combination of these parameters, and to obtain reasonable clusters, we tested three $R_{max}$ values equal to 100, 200 and 300$km$, and three $T_{max}$ values equal to 1, 3 and 5 years. The choice for $R_{max}$ is corroborated by Zhang *et al.* (2010) who report measurements constantly less than 500 $km$ for the radius of typical convective storms in the Chinese mainland, which can trigger flash floods. Results of STPSS for each of the nine combinations of these parameters are shown in Figure 4.


Table 2: Number of detected spatiotemporal clusters of flash flood disasters in China during
1950-2015 using different parameters.

| $R_{max}$ (km) | $T_{max}$ (years) | | |
|---|---|---|---|
| | 1 | 3 | 5 |
| 100 | 131 | 128 | 130 |
| 200 | 85 | 77 | 75 |
| 300 | 58 | 54 | 53 |

The largest variation in the number of detected clusters is mainly associated with $R_{max}$ –
as $R_{max}$ increases, the detected flash flood disaster clusters exhibit a clear decrease – rather
than with $T_{max}$. This result is summarized in Table 2. More specifically, large $R_{max}$ values
affect the detection of clusters acting at a fine scale, which tend to be missed or merged into
larger ones; conversely, very large clusters, acting at a coarse spatial scale, are still detected.
This is geographically visible in the south-easternmost sector of China (Figure 4). Changes
on $T_{max}$ have almost no effect on the number of clusters since, even allowing for a maximum
duration of 5 years, almost all the clusters do not exceed one year. As complementary
information, Table 3 presents the temporal duration, expresses as start and end date, for the
first ten clusters of flash flood disasters using $T_{max}$ equals to 3 years and for increasing values
of $R_{max}$, equal to 100, 200 and 300 $km$. Results confirm that the cluster duration does not
exceed one year. The most significant cluster was detected in 1975, while the rating for the
following clusters can change in the three cases. Nevertheless, it is important to notice that
the top-ten clusters are well distributed over the entire study period, with the oldest one
detected between 1963 and 1969.

### 3.2.2 Clusters characterization

Detected clusters where further analyzed by considering the impact of flash flood disasters.
To this end, we examined only clusters detected by using $R_{max} = 200km$ and $T_{max} = 3years$.
The choice of a $T_{agg} = 1year$ was originally meant to focus our analyses on effects that may
exhibit a yearly cycle. However, this would have smoothed nested effects acting at the
seasonal scale. For this reason, we opted to carry out additional analyses using a $T_{agg}$ of
three months (hereafter referred as *monthly model*). Results are shown in Figure 5 where
information on the spatial distribution of the detected clusters is combined with the impact
related to the single flash flood events (see Table 1). Overall, the clusters chiefly appear
along the main river systems in China, namely the Yangtze, the Yellow, the Pearl and the
Yarlung Zangbo Rivers. In addition, some clusters stand out on high mountains such as the
Qinling-Daba and the Changbai Mountains.

Forcing the model parameterization to aggregate the time over a fraction of the year (three
months) allows us to investigate potential seasonal effects. Indeed, even if the maximum
temporal duration is still of one year, looking at the ten most significant clusters detected



Table 3: Temporal duration of the first 10 clusters of flash flood disasters detected via three different models (left: $R_{max} = 100km$; center: $R_{max} = 200km$; right: $R_{max} = 300km$)

| ID | Radius | Start date | End date | ID | Radius | Start date | End date | ID | Radius | Start date | End date |
|----|--------|-----------|----------|----|--------|-----------|----------|----|--------|-----------|----------|
| 1 | 81.04 | 1975/1 | 1975/12 | 1 | 81.04 | 1975/1 | 1975/12 | 1 | 81.04 | 1975/1 | 1975/12 |
| 2 | 64.51 | 2010/1 | 2010/12 | 2 | 146.06 | 1998/1 | 1998/12 | 2 | 146.06 | 1998/1 | 1998/12 |
| 3 | 60.73 | 2006/1 | 2006/12 | 3 | 64.51 | 2010/1 | 2010/12 | 3 | 64.51 | 2010/1 | 2010/12 |
| 4 | 72.76 | 2010/1 | 2010/12 | 4 | 60.73 | 2006/1 | 2006/12 | 4 | 60.73 | 2006/1 | 2006/12 |
| 5 | 94.42 | 1998/1 | 1998/12 | 5 | 72.76 | 2010/1 | 2010/12 | 5 | 72.76 | 2010/1 | 2010/12 |
| 6 | 73.13 | 1969/1 | 1969/12 | 6 | 73.13 | 1969/1 | 1969/12 | 6 | 73.13 | 1969/1 | 1969/12 |
| 7 | 56.67 | 1963/1 | 1963/12 | 7 | 176.96 | 1982/1 | 1982/12 | 7 | 176.96 | 1982/1 | 1982/12 |
| 8 | 49.51 | 1996/1 | 1996/12 | 8 | 70.57 | 1984/1 | 1984/12 | 8 | 70.57 | 1984/1 | 1984/12 |
| 9 | 70.57 | 1984/1 | 1984/12 | 9 | 129.06 | 1996/1 | 1996/12 | 9 | 129.06 | 1996/1 | 1996/12 |
| 10 | 35.27 | 1987/1 | 1987/12 | 10 | 157.14 | 2010/1 | 2010/12 | 10 | 157.14 | 2010/1 | 2010/12 |

Table 4: Temporal duration of the first 10 clusters of flash flood disasters during 1950-2015 ($R_{max} = 200km$, $T_{max} = 1year$, $T_{agg} = 3months$).

| ID | Radius | Start date | End date |
|----|--------|-----------|----------|
| 1 | 54.88 | 2010/10/1 | 2010/12/31 |
| 2 | 81.04 | 1975/4/1 | 1975/9/30 |
| 3 | 72.76 | 2010/7/1 | 2010/9/30 |
| 4 | 146.06 | 1998/4/1 | 1998/9/30 |
| 5 | 60.73 | 2006/7/1 | 2006/9/30 |
| 6 | 73.13 | 1969/4/1 | 1969/9/30 |
| 7 | 178.05 | 1982/7/1 | 1982/9/30 |
| 8 | 199.88 | 1996/4/1 | 1996/6/30 |
| 9 | 157.14 | 2010/7/1 | 2010/9/30 |
| 10 | 67.05 | 1984/4/1 | 1984/9/30 |
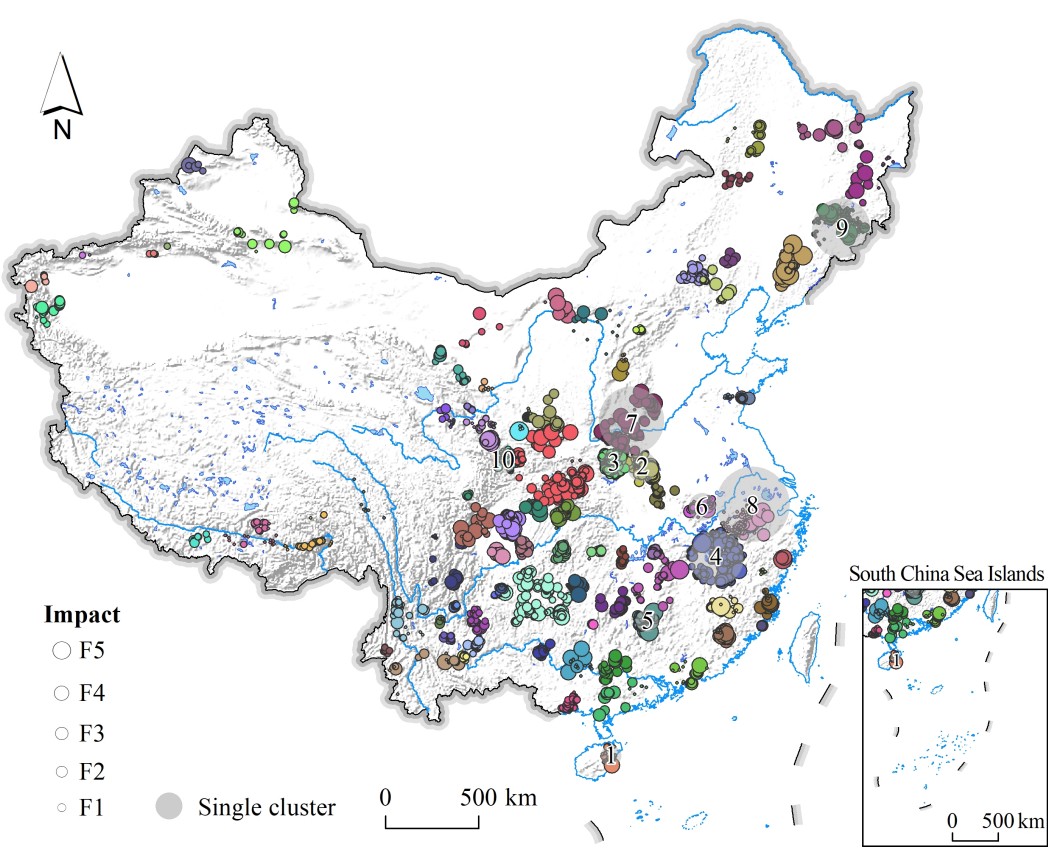

Figure 5: Significant (p<0.005) spatiotemporal clusters of flash flood disasters in China during 1950-2015 ($R_{max} = 200km$, $T_{max} = 3years$, $T_{agg} = 3months$). Each event belonging to a single cluster is further resized as a function of its impact, in accordance to Table 1.




under the *monthly model* (Table 4), it results that all of them have a duration of three (six clusters) or six (four clusters) months. Notably, almost every cluster (nine clusters) encompass the period from July to September, with an earlier start date (in April) for the ones which have a longer duration.

### 3.2.3  Temporal duration of detected clusters

The temporal variation in the duration of the detected clusters could have been driven by the precipitation regime. In additional, spatiotemporal dependency may have been induced by the geomorphological setting of the area and by anthropogenic pressures, but these last factors should have a minor effect compared to the rainfall pattern, which acts as the primary triggering factor of flash floods. Therefore, in the present study we assume the precipitation as the main driver for flash floods detected clusters, and results are interpreted and discussed on the basis of this hypothesis. Allowing for $T_{max} = 3 years$ in the parameterization of the *yearly models*, the temporal duration of the detected clusters ranges from one to three years (see Figure 6). The cluster detection pattern appears quite clear and well defined. However, since 1980 some clusters partially overlap. This can lead to two separate interpretations. Firstly, the relative small number of clusters detected between 1950 and 1980 may imply that the data acquisition and report in the Chinese database of hydro-morphological disasters was not fully operational at the time. Conversely, from 1980 to present days the Chinese database has evolved into a mature and detailed geographic information system. Secondly, the same pattern can be justified as a result of climatic changes. In fact, overlapping clusters of one, two and three years duration essentially appear only after 1980. These concurrent clusters may reflect similar synchronous variations of the climate settings and rainfall regimes across China in the recent period.




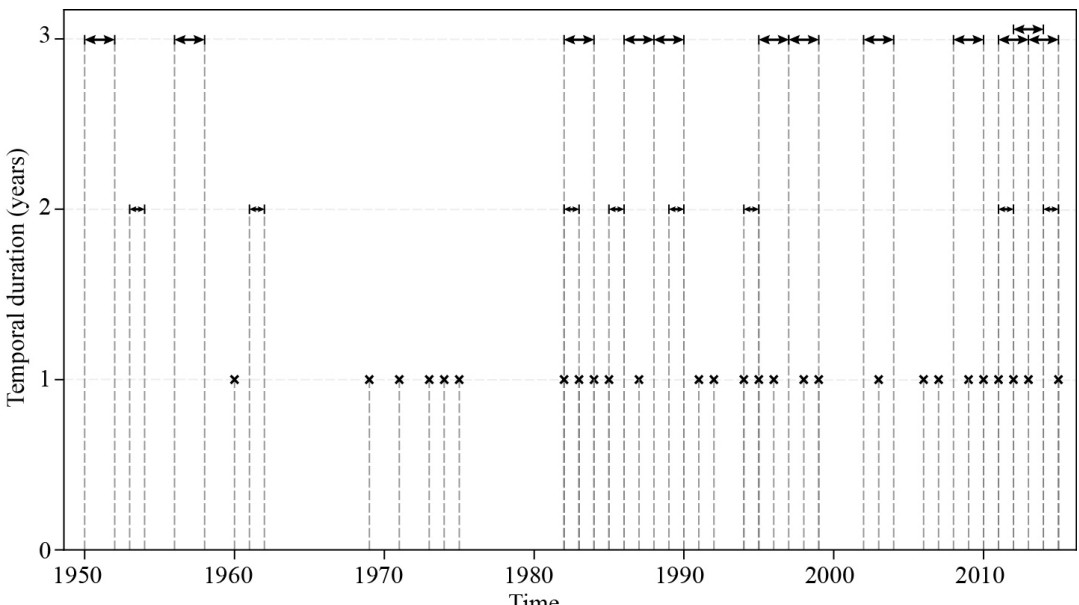

Figure 6: Temporal duration of flash flood disasters clusters in China during 1950-2015 ($R_{max} = 200km$, $T_{agg} = 1year$, $T_{max} = 3years$).

We summarized the same results for the *monthly model* in Figure 7. To better visualize the seasonality trend, we opted for a cyclic representation of the detected clusters, plotting their pattern in four temporal duration classes of 3, 6 and 9 months as well as one year. Most clusters show a 3-months duration, concentrated in the period between July and October, and an increasing density after 1980. Furthermore, clusters of 6-months temporal duration are most likely to occur from January to July or from April to October. As for clusters with 9-months temporal duration, these mostly cover the period of July-August-September, irrespective of the starting month. Ultimately, as noticed for the *yearly model*, also in the *monthly model* much more clusters were detected in the late period, mainly from 2000. Moreover, the vast majority of flash flood disasters clusters happened between July and October, a period coinciding with the wet season in China.

### 3.2.4 Recurrence of clusters at decades-scale

The analyses run in the previous sections were all voted to search for clusters in a relatively small temporal window. However, environmental changes, and especially those related to climate change, usually act on a longer time-span. To better investigate this effect, we considered a temporal subdivision of the dataset into six subsets, each one lasting ten years (starting from 1956). For each decade (1956-1965, 1966-1975, 1976-1985, 1986-1995, 1996-2005, 2006-2015) the following parameter for the scanning widows were imposed: $R_{max} = 200km$, $T_{max} = 2years$ and $T_{agg} = 1year$. As shown in Figure 8, the number of detected clusters

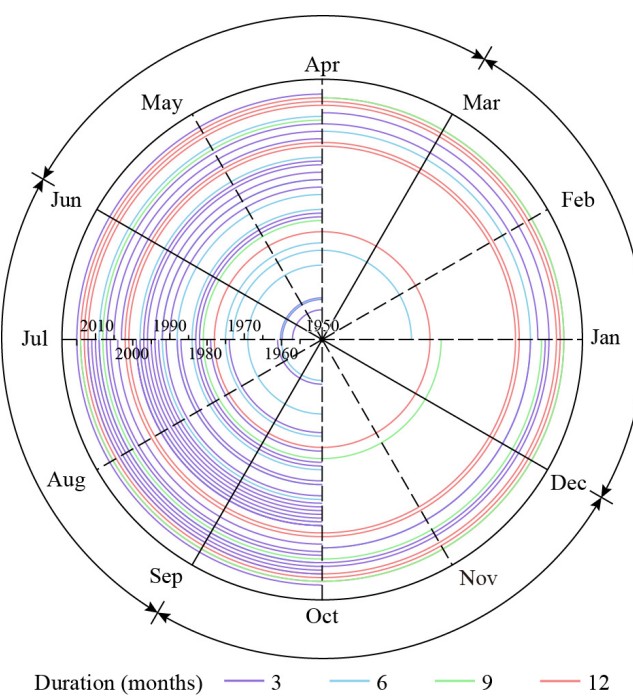

Figure 7: Seasonal effect of flash flood disasters clusters in China during 1950-2015 ($R_{max} = 200km$, $T_{max} = 1year$, $T_{agg} = 3months$).

increases from the early to recent periods. These are compared with the rainfall distribution, derived from the daily rainfall data provided by the China Meteorological Administration (http://data.cma.cn/). In the present study, only the weather stations (a total of 699 rain gauges) with complete data for the period 1955-2015 were considered. The mean monthly and annual rainfall were computed for each station and this data were then regionalized on a $2km \times 2km$ lattice, via Ordinary Kriging interpolation. It results that flash floods detected clusters are mainly located in the southeastern most humid regions in every period. However, in the last two decades, clusters appear also in the northwestern arid regions. Even if the rainfall distribution, averaged over each decades, does not allow to discover clear changes along the subsequent periods, these newly detected clusters can be due to the intensification of the extreme rainfall events occurring in the area in recent periods. This assumption is confirmed by the statistics on clusters duration (Figure 9). From the boxplot summarizing the descriptive statistics it is evident that the median values of clusters duration tends to slightly decrease from 46 days (1956-1965), to 17 days (1986-1995), to stabilise at a value around 20 days in the two last decades. At the same time, the overall duration, measured as difference between the maximum and the minimum value, is higher in the late periods (140 days in 1956-1965, and 93 and 74 days respectively in the two following decades) than in the early periods (about 65 days for the last two decades). This is even more evident

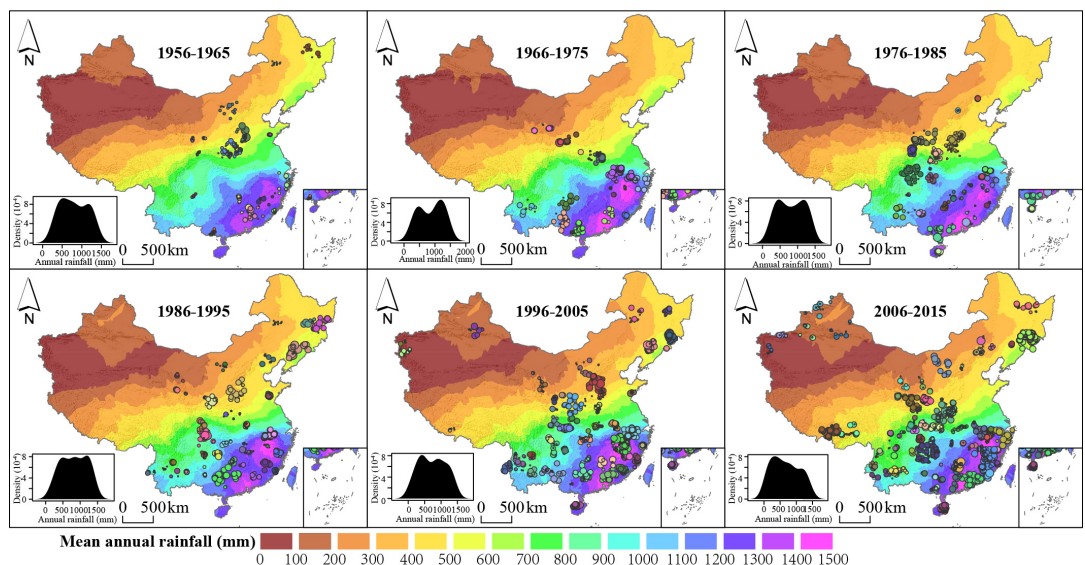

Figure 8: Significant (p < 0.005) spatiotemporal clusters of flash flood disasters in China every ten years. The size of the circles indicates the impact of flash flood disasters according to the classification proposed in Table 1.

354  looking at the inter quantile ranges, which decrease with time. To resume, from these analy-
355  ses, the number of detected clusters globally increase in time, but their duration drastically
356  decreases in the recent period, indicating a possible activation induced by short-duration
357  extreme rainfall events.

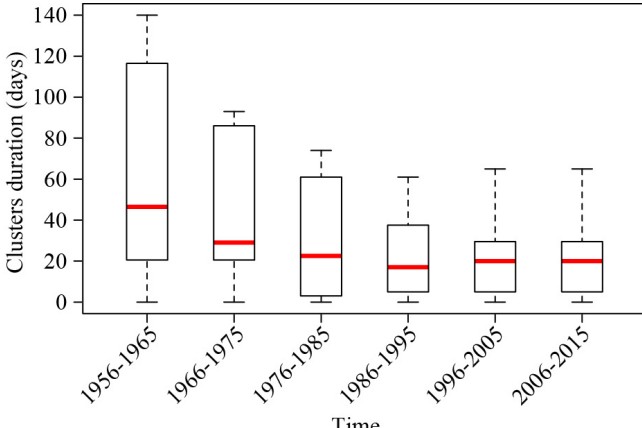

Figure 9: Boxplots summarizing the descriptive statistics of the duration of clusters reported on Figure 8.
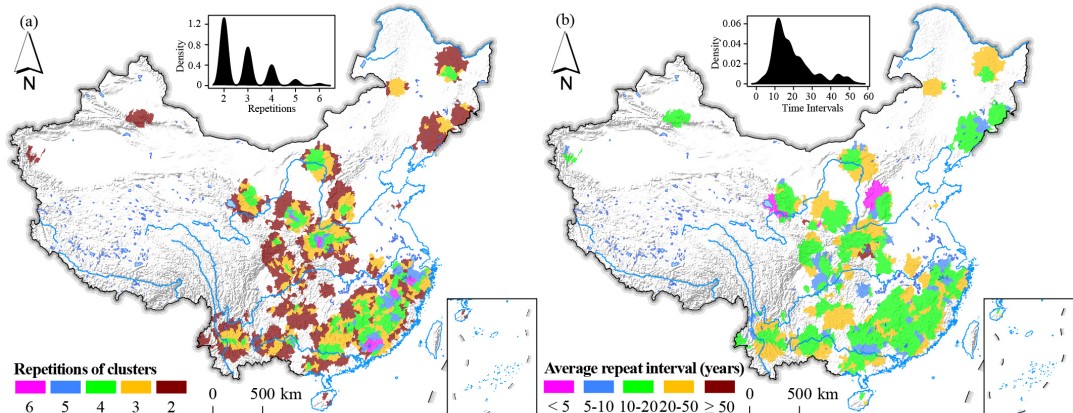

Figure 10: Catchments with clusters detected more than once (a) and return period for the clusters (b).

We further explored how many times the clusters detected from the previous investigation overlap, considering the catchment level. Results provide an useful information on the recurrence of clusters of flash flood disasters every ten years. To perform this analysis, the centroid of each cluster (with reference to Figure 8) was extracted and intersected with the catchment boundaries. In a second step, the number of repeated clusters per catchment was computed and their distribution investigated through the time. Results are shown in Figure 10, where panel (a) reports the number of repeated clusters and panel (b) reports the information on their relative occurrence across time (similarly to the concept of return time but in the context of spatiotemporal clustering).

Figure 10a shows that, as for the spatial trends of the detected clusters, the catchments with recurrent clusters are mainly located in the southeast sector and essentially in the coastal mountains. From Figure 10b it emerges that, on average, most of the repeated cluster occur with an interval between 10-20 and 20-50 years.

# 4   Discussions

The present study aims at exploring the spatiotemporal clustering characteristics of flash flood disasters in China. For this purpose, we analyzed the official historical inventory, which covers a very long period (from 1950 to 2015). Results are interpreted with a particular regard to the rainfall distribution, being these two processes highly related (Wei *et al.*, 2018). The spatiotemporal K-function was fist computed to assess the deviation of flash flood pattern distribution from a random process. This revealed a clustering behavior at specific spatial distances and yearly periods. Scan Statistics, the spatiotemporal permutation model we adopted, was then performed to identify statistically significant clusters together with their duration (start and end date). This allowed us to detect areas and periods more susceptible



to flash flood disasters. We opted for a set of possible combinations for the maximum spatial and temporal extension of the scanning windows, while the data were aggregated both at yearly and at seasonal scale. More specifically, we tested three $R_{max}$ values equal to 100, 200 and 300 $km$, and three $T_{max}$ values equal to 1, 3 and 5 years for the *yearly model*, with an aggregation of three months for the *monthly model*. The most significant cluster resulting from the yearly model was detected in 1975, while the rating for the following clusters can change by varying $R_{max}$; nevertheless, it is important to note that the top-ten clusters are well distributed over the entire study period, with the oldest one detected in 1963-1969. Results of the monthly model show that the top-ten detected clusters have a duration of three (six clusters) or six (four clusters) months. Notably, almost every cluster encompasses the period from July to September, a period coinciding with the wet season in China, with an earlier start date (in April) for the ones which have a longer duration. Globally, much more clusters were detected in the late period, mainly from 2000. Overall, clusters are chiefly located along the main river systems in China (the Yangtze, the Yellow, the Pearl and the Yarlung Zangbo Rivers). In addition, some clusters stand out on high mountains such as the Qinling-Daba and the Changbai Mountains.

Finally, to detect changes acting at a larger temporal scale, dates were grouped each ten years over the last six decades (from 1956 to 2015). As for the previous analyses, detected clusters are mainly located in the southeastern most humid regions in every period. However, in the last two decades, clusters appear also in the northwestern arid regions. These newly detected clusters can be due to the intensification the extreme rainfall events occurring in the area in recent periods, as a consequence of climate changes (Song *et al.*, 2011a). This important fact is confirmed by checking the descriptive statistics of the duration of clusters: globally, the number of detected clusters increases in time, but the duration drastically decreases in recent periods, indicating a possible activation induced by short-duration extreme rainfall events. Our analyses reveled that the catchments with recurrent clusters are mainly located in the southeast sector and essentially in the coastal mountains. China is indicated as one of the hotspot with global flood-exposed coastal population (Van Coppenolle and Temmerman, 2020). Therefore, we can assume these catchments to be exposed at the highest potential risk across the whole Chinese territory also in the short to long term future. Nevertheless, catchments with repeated clusters in a shorter time-span (5 to 10 years) may also pose a relevant threat, especially in the near future.

In the present study spatiotemporal clusters of flash floods were detected chiefly on the basis of two parameters ($R_{max}$ and $T_{max}$), without featuring terrain attributes, precipitation regimes and anthropogenic pressure. However, these factors may have played and still play a significant role to explain the distribution of flash flood disasters. For instance, the approach we adopted may over-rely on spatial distances to detect clusters. In fact, the natural landscape has mountain belts that can act as orographic barriers to the incoming cloudbursts, effectively limiting the rainfall distribution – hence flash flood occurrences – on one or the other side of a catchment divide (at various scales). As for the temporal scale, due to the




large time-span, the detected temporal patterns may reflect more information due to long-term climatic variations rather than specific conditions. For this reason, we are planning to extend our spatiotemporal cluster analyses to more complex models, which can concurrently capture multivariate contributions featuring environmental effects, even at the latent level (Lombardo *et al.*, 2018, 2019a).

# 5 Conclusion

In this work, we explore the national archive of flash flood disasters in China from 1950 to 2015. The term disaster is meant to describe the destructiveness of the flash floods, since each record in this archive has produced economic, life losses, or both.

The clustering procedure highlighted distinct spatial and temporal patterns at different scales. For instance, flash flood disasters cluster in specific regions and closely follow the mean rainfall distribution. Additionally, we were also able to distinguish seasonal, yearly and even long-term flash flood persisting behaviors. The persistence of disasters is a crucial information because it indicates the risk that a community may undergo in response to a flash flood. Moreover, we studied the cycle of such disasters with particular emphasis on their repeated occurrence per catchment. This complementary information can be further used in relation to engineering and structural design. In fact, infrastructure is usually built to sustain the damage of an event of certain return time. In our analyses, we show that the very same area has been hit and incurred losses up to six times in the last 66 years. This may suggest locally-tailored structural improvements which may lengthen the life expectancy of specific infrastructure as well as reduce the number of victims.

We would like to stress that, as advanced as it may be, our clustering framework is essentially a descriptive tool. And yet, the amount of information one can draw from a descriptive tool can be extremely valuable. Nowadays, the hazard community's effort is mainly dedicated to predictive modeling of various natures and purposes, thus leaving under-explored or even unexplored some basic concepts and interpretative conclusions that data description and visualization can provide. Long time series of national hazard phenomena are one of these examples where studying variations over space and time can highlight very important environmental dynamics, even in the direction of climate change and its implications.

# Acknowledgement

This work was supported by the China National Flash Flood Disasters Prevention and Control Project. The authors are grateful for financial support from the China Institute of Water Resources and Hydropower Research (IWHR), grant number No. SHZH-IWHR-57 and National Natural Science Foundation of China, grant number No. 41571388.





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
