# Peer review of "Spatiotemporal clustering of flash floods in a changing climate (China, 1950-2015)"

_Natural Hazards and Earth System Sciences, 2020_

## Referee Comment (RC1) · Anonymous Referee #1 · 23 Oct 2020

**Review for Natural Hazards and Earth System Sciences (NHESS)**

**Spatiotemporal clustering of flash floods in a changing climate (China, 1950-2015)**

**Nan Wang, Luigi Lombardo, Marj Tonini, Weiming Cheng, Liang Guo7, Junnan Xiong**

This manuscript investigates the long-term flash floods persistency over China based on comprehensive catalogue data via Ripley's K-function and the Scan Statistics. They claim that the principal mechanism or triggering factor that controls the spatial and temporal distribution of flash flood is rainfall and rainfall characteristics. On the other hand, they claim that flash flood characteristics like duration, etc. are also controlled by rainfall with changing climate (ie. Climate change). This paper has strong data and rigorous analyses to investigate flash flood phenomena. However, there are some missing perspectives to explain the control mechanism of flash floods. Based on this I would be in favour of publication of the paper however, I have the following points to raise before a possible publication is granted:

- The basin morphometric characteristics are ignored in this study because precipitation is considered a major factor. However, many geomorphometric parameters have an important role in flash flood characteristics. For instance, the time of concertation is quite important for the flood, and the shape of the basin controls these parameters. When the water has reached to surface (basin), morphometric parameters control the way of water. On the other hand, flash floods are mentioned as hydro-geomorphological hazard almost in every sentence, but it is claimed that there is no any impact of "geo". How it can be a small impact than rainfall? It wouald be better explained in the paper.

- The impact of land-use on flash floods has ignored as well even if they claim that these parameters are not as important as rainfall. However, Yang and Tian (Abrupt change of runoff and its major driving factors in Haihe River Catchment) has written a manuscript on abrupt changes in runoff. They expressed that human activities have a strong impact on runoff changes rather than climatic changes because of Chinese land reform. Therefore, many studies like mentioned above showed that land-use changes have an important effect on surface water, infiltration, etc. For a very dynamic country like China, How can land-use be unimportant parameters? Of course each storm can cause flash flood but also land use changes have important impact by chancing infiltration capacity. As a humble suggestion, I suggest that authors can match the important cluster dates with important land-use changes in China. If they still think that

land-use has no importance on flash floods in china they have to explain why. Otherwise, aspect of this study will be neglected.

- L105-115: The aim of this study should explain more clearly. What is the reason that study has done? What does this study bring new concept to flood studies etc.? What is the novelty of this study? I understand that this is first attempt to flood characteristic with such a kind comprehensive data. These kind of questions should explain more clearly.

- L158: I was wondering that how the past economic losses due to flash floods have converted current currency in the study.  As far as I understand, this is one of the core of perception of this study in order to distinguish the impact of flash floods in China

- L218-224: How were these gaps betwee distances determined? There are some distances used for Ripley K function but there is no any explanation why this distances were used. Based on this K function, authors have determined the Rmax as well. Also, aggregation numbers seems similar like K function. Are they arbitrary numbers? Therefore, it would better to explain clearly.

- L341-343: "It results that flash floods detected clusters are mainly located in the southeastern most humid regions in every period. However, in the last two decades, clusters appear also in the north-western arid regions". It is not surprising to see the south areas precipitation are coinciding with important cluster dates. Because these areas correspond to the most important tropical and subtropical cyclone areas. I think authors are missing this perspective. Hu et al., has conducted research about flood mortality for the world (https://doi.org/10.1016/j.scitotenv.2018.06.197). They have explained the reason why tropical cyclones have an important impact on flood-induced mortality in those areas. Therefore, it is not surprising to see time patterns in Figure 7. Authors should also consider this perspective in their study. For instance, is there any relations between cyclone frequency and flash floods? Of course rainfall is an output of cyclonic storm but the cyclone itself is the mechanism of precipitation and flood formation.

- Authors have mentioned about storms but gave annual mean rainfall in Figure8. Does it make sense this kind of rainfall for flash-floods? On the other hand, this mean rainfall has obtained from sum of monthly rainfall. For flash floods, does it make sense as well? I would suggest to authors to use the long-term mean of 95 percentile of daily rainfall. May be, they can find more interesting results. ERA dataset also can be used for that comparison.

- L398-402: Sometimes it is easy to blame the climate change for some catastrophic disasters. However, it was not seen any analysis in this study for climatic changes, even extreme rainfalls. Therefore, I think that it would be valuable to investigate the rainfall characteristics in each spatial and time clusters. Therefore, I again suggest that, daily rainfall characteristics such as 95 percentiles, 5-day max rainfall can be investigated in each clusters. However, if there is no data to investigate this phenomena, I would suggest open access datasets.

---

## Referee Comment (RC2) · Anonymous Referee #2 · 9 Nov 2020

This work focuses on spatiotemporal cluster analysis of flash flood disasters in China with the overall goal, as claimed by the authors, to identify and characterize spatiotemporal dependencies of flash flood related disasters. The scientific objectives of the manuscript are highly relevant to the scope of this journal and of interest to its readership. However, in my opinion, the current version of the manuscript does not succeed to deliver scientific findings in a clear and convincing way. I provide below my major and minor comments that will hopefully help the authors to improve their work.

Major comments:

1. How can we be sure that part of the findings is not related to the way the database was constructed? For example, can the duration of clusters shown as boxplots in Figure 9 be a result of infrequent records before 1970?

2. More effort needs to be done to link findings on the clusters with the physical meaning of flash flood related properties or occurrence. P19,L385 authors state "The most significant cluster resulting from the yearly model was detected in 1975", so what does this mean exactly? You need to improve a lot the interpretation of results in the current manuscript and help the reader understand what the different findings actually mean. Otherwise the work will remain predominantly a cluster analysis with little information on the characteristics of the actual hazard.

3. As a follow up of previous point, you present a number of findings (in terms of cluster attributes etc) that add more confusion than clarity. Cluster characteristics for different temporal thresholds of 1,3,5 yrs or monthly models etc are provided but interpretation/significance of each of those is not clearly delivered.

4. P16,L345 "these newly detected clusters can be due to the intensification of the extreme rainfall. . ..". This can potentially be a very interesting point, but more work is needed to justify this. You would have to actually look at the precipitation record in the area and do an event-based analysis to find whether indeed extreme storm event are more frequent.

5. Definitions/explanation of certain terms and approaches is required. How is the term "repeated clusters" defined? How is "their relative occurrence (similarly to the concept of return period. . .)" determined?

6. How do you define "impact" in the context of events selected? I assume that impact is somehow related to exposure and therefore all the events analyzed are related to areas where there is exposure. So this does not mean that in areas not shown on your map, there were no flash flood events occurring. I think this should be clarified to highlight once again that the analysis is highly related to the existing database.

7. Conclusions: need to be improved to provide more quantitative information on the overall findings. For example, elaborate more on what you found on the repeated occurrence per catchment.

Minor comments:

1. For Figure 1 and other similar figures: results for the islands are not visible at all. Also what does the dash line represents? My understanding is that certain island territories are under dispute among several countries so I would suggest caution on how your represent those.

2. Fig.4: a legend to explain colors is missing.

---

## Author Comment (AC1) · 20 Dec 2020

**Reviewer 1**

**General comments:**

This manuscript investigates the long-term flash floods persistency over China based on comprehensive catalogue data via Ripley's K-function and the Scan Statistics. They claim that the principal mechanism or triggering factor that controls the spatial and temporal distribution of flash flood is rainfall and rainfall characteristics. On the other hand, they claim that flash flood characteristics like duration, etc. are also controlled by rainfall with changing climate (i.e. Climate change). This paper has strong data and rigorous analyses to investigate flash flood phenomena. However, there are some missing perspectives to explain the control mechanism of flash floods. Based on this I would be in favour of publication of the paper however, I have the following points to raise before a possible publication is granted:

We would like to thank the Reviewers for their work and valuable comments, which have been constructive and useful to improve the quality of the manuscript. We provide in the following rebuttal a point-by-point response to your the specific comments.

**Specific comments:**

1. The basin morphometric characteristics are ignored in this study because precipitation is considered a major factor. However, many geomorphometric parameters have an important role in flash flood characteristics. For instance, the time of concentration is quite important for the flood, and the shape of the basin controls these parameters. When the water has reached to surface (basin), morphometric parameters control the way of water. On the other hand, flash floods are mentioned as hydrogeomorphological hazard almost in every sentence, but it is claimed that there is no any impact of "geo". How it can be a small impact than rainfall? It would be better explained in the paper.

We thanks the reviewer for this relevant comment. Indeed, geomorphometric characteristics are one of the main factors considered for flash flood susceptibility and risk mapping, assuming that the overall settings do not dramatically change over time. However, in the present study, we are considering both the spatial AND the temporal domains with the aim of detecting spatiotemporal clusters occurring as a consequence of the interaction between these two domains (see L.94-100; L.115-120). Therefore, clusters are influenced by dynamic factors, which continuously vary both in space and time, such as rainfall. Static factors (such as geomorphometry) do not significantly change with time (during the investigated period, or at least to an extent that we cannot measure at the scale of the whole Chinese country). They could explain the spatial pattern of clusters if we were looking at this dimension only. But, in our case clusters are detected if and only if flash flood events closer in space are ALSO closer in time. Accordingly, their occurrence, in terms of clusters spatiotemporal distribution and cluster duration, is compared only with the rainfall because this is the only parameter that covers and varies across the same spatiotemporal domain, as the clusters

themselves. Nevertheless, to make this concept much more clear for the NHESS readership, following your suggestions, we have now added a thorough explanation in the discussions (see L.384-392).

2. The impact of land-use on flash floods has ignored as well even if they claim that these parameters are not as important as rainfall. However, Yang and Tian (Abrupt change of runoff and its major driving factors in Haihe River Catchment) has written a manuscript on abrupt changes in runoff. They expressed that human activities have a strong impact on runoff changes rather than climatic changes because of Chinese land reform. Therefore, many studies like mentioned above showed that land-use changes have an important effect on surface water, infiltration, etc. For a very dynamic country like China, how can land-use be unimportant parameters? Of course, each storm can cause flash flood but also land use changes have important impact by chancing infiltration capacity. As a humble suggestion, I suggest that authors can match the important cluster dates with important land-use changes in China. If they still think that land-use has no importance on flash floods in china, they will have to explain why. Otherwise, aspect of this study will be neglected.

Following the suggestion of the reviewer, we looked for quantitative data related with land use changes in China, which could help us to find a match with respect to detected clusters, similarly to what we have done in Fig.8 with rainfall data. Despite China experienced land use and land cover changes during the investigated period – as also confirmed by the vast literature we referred in the manuscript – to the best of our knowledge, a quantitative study investigating the entire country over entire study period (1955- 2015) has never been carried out up to now. Therefore, we cannot get access to land use change data over the same spatiotemporal domain covered by the flash floods we studied here. Also, it is out of the scope of the present study to produce accurate maps of land use changes in China ourselves.
The best we could do was to look at population density maps, freely available at http://worldmap.harvard.edu/chinamap/. These attest for an increase of the population density from 1953 to 2010 in the more urbanised southeastern area, confirmed by the global land cover maps
(http://maps.elie.ucl.ac.be/CCI/viewer/index.php). To make things worse, this information is available only from 1992 up to present days, thus it covers only one third of the spatiotemporal domain under study and it does not correspond to the land use characteristics mentioned by Rev1.
To resume and point out these findings, we have now added a new paragraph in the discussions. There, we consider the possible influence of human activities in the spatiotemporal dynamic of detected clusters. Several citation, included Yang and Tian, were added accordingly (L.384-392).

3. L.105-115: The aim of this study should explain more clearly. What is the reason that study has done? What does this study bring new concept to flood studies etc.? What is the novelty of this study? I understand that this is first attempt to flood characteristic with such a kind comprehensive data. These kind of question should explain more clearly.

Thank you for pointing this out. We have now provided a more clear explanation of the aim and novelty of the present study, as per your suggestion. Specifically, to this end, we have restructured the last paragraph of the introduction, which we have modified as follows:

(L.101-120) "Therefore, it is especially useful to investigate large spatiotemporal inventories of hydro- and geo-morphological processes such as flash floods. Indeed, the detection of clusters originated by events closer both in space and in time can be more informative than the simple investigation of their purely temporal and purely spatial pattern distribution. For example, understanding the duration of the spatiotemporal clusters of flash floods is key tool to investigate their dynamic and to highlight more vulnerable area and frame period.

In light of this, the main objective of the present research is to explore the pattern distribution of flash flood disaster which have caused life and/or economic losses in China over a 65-years period (daily data from 1950 to 2015). Firstly, the Ripley's K-function was applied to explore the deviation of flash flood disasters from a random process. Results allows to assess at which spatial and temporal scales events are clustered. Then, a local cluster indicator, namely Scan Statistics, was implemented to map statistically significant spatiotemporal clusters. To the best of our knowledge, this study represents the first attempt of investigating the spatiotemporal cluster behaviour of flash flood disaster affecting a huge area, such as the entire Chinese territory. Moreover, the volume of the data that we analyzed represents an additional challenge allowing to provide useful insights on flood dynamics over a large spatiotemporal domain and enabling considerations in the context of climatic changes. To this end, we finally compared the dynamic of the clusters, detected from the early to the recent period, with the mean rainfall evolution, computed each 10-years, which is assumed as a local climatic proxy factors"

4. L.158: I was wondering that how the past economic losses due to flash floods have converted current currency in the study. As far as I understand, this is one of the core of perception of this study in order to distinguish the impact of flash floods in China.

Thank you again, we realized this was something that both reviewers have pointed out to be a problematic content because of various reasons we did not originally consider. For instance, the exchange rate between RMB and USD, as well as the inflation of the currency through time. We actually looked for this information during the revision process but we could only find it starting from the late 80'ies onwards. Because of this, and in consideration of the comments made by both the reviewers, we finally decided to remove from the paper any reference to the impact of flash flood disasters (Table 1) and to modify accordingly Fig.5 and Fig.8, where clusters are represented with a size that is proportional to the impact classes. This because these results can add more confusion than clarity. We genuinely thank you for pointing this out, as we realized adding this information was not essential and was actually making our message not just more unclear but also misleading in some way.

5. L.218-224: How were these gaps between distances determined? There are some distances used for Ripley K function but there is no any explanation why these distances were used. Based on this K function, authors have determined the Rmax as well.

Also, aggregation numbers seem similar like K function. Are they arbitrary numbers? Therefore, it would better to explain clearly.

Indeed the main insight from the 3D-plot is the global spatiotemporal pattern behaviour of the flash flood, which results to be clustered at all distances. Nevertheless it seems to be hard to provide precise values. Therefore, for a better understanding of the 3D-plot, we reformulated the paragraph (Results' section) as follows:
(L.226-241) "In the present study the spatiotemporal K-function was used to assess the global cluster behavior of flash flood disasters generated by the interaction between these two variables. To this end, the perspective 3D-plot of D(s,t) represents a useful visual tool allowing to estimate the distribution pattern of events along the spatial and the temporal dimension. In more details, positive values attest for a cluster distribution, while values close to zero indicate a random pattern, with no interaction between space and time. In our case, the 3D-plot (Figure 2) shows that at any distance, from hundreds to thousands meters, and from few years to decades, flash flood events display a cluster behaviour, which is more pronounced at increasing distance-values. In addition, the spatiotemporal K-function was computed considering individually the southeastern and the northwestern area in China, given that the first corresponds to the rainiest zone, highly affected by flash floods, while the second is predominantly desert. It results that (Figure 3) in the southeastern China (panel a) clusters arise at a shorter spatial distance and closer in time than in the northwestern China (panel b). As regards the temporal dimension, the two areas show a similar cluster behaviour, with a strong attraction among events up to 10-years, and than lasting in time with a more relaxed clustering behaviour."

6. (L.341-343) "It results that flash floods detected clusters are mainly located in the southeastern most humid regions in every period. However, in the last two decades, clusters appear also in the north-western arid regions". It is not surprising to see the south areas precipitation are coinciding with important cluster dates. Because these areas correspond to the most important tropical and subtropical cyclone areas. I think authors are missing this perspective. Hu et al., has conducted research about flood mortality for the world (https://doi.org/10.1016/j.scitotenv.2018.06.197). They have explained the reason why tropical cyclones have an important impact on flood-induced mortality in those areas. Therefore, it is not surprising to see time patterns in Fig.7. Authors should also consider this perspective in their study. For instance, is there any relations between cyclone frequency and flash floods? Of course, rainfall is an output of cyclonic storm but the cyclone itself is the mechanism of precipitation and flood formation.

Actually, to provide strong and quantitative evidence of correlation between cyclone frequency and flash floods a statistical correlation analysis should be carried out. Although this is out of the scope of the present study, following the indications of the reviewer, authors added some relevant consideration concerning this prospective in the Discussions (see L.426-435).

7. Authors have mentioned about storms but gave annual mean rainfall in Figure8. Does it make sense this kind of rainfall for flash-floods? On the other hand, this mean

rainfall has obtained from sum of monthly rainfall. For flash floods, does it make sense as well? I would suggest to authors to use the long-term mean of 95 percentile of daily rainfall. May be, they can find more interesting results. ERA dataset also can be used for that comparison.

Thank you for your feedback once again. This was a brilliant suggestion indeed. To follow your suggestion we have performed the same analyses one again but instead of using the mean rainfall we have now introduced a new term corresponding to the extreme precipitation. Specifically, we have computed what follows:

- For each weather station, we have extract the time series of the daily rainfall records.
- From each time series, we have extracted a subset corresponding to values greater than the $95^{th}$ percentile, obtaining a new and smaller vector of rainfall extreme values.
- For each weather station, we have taken the cumulative rainfall out of the extreme subset, on a yearly basis. This operation produces 10 aggregated extreme values per decade.
- We have then extracted the mean extreme value out of the 10 mentioned above to be the representative extreme rainfall per rain gauge and per decade.
- Finally, we have interpolated these values over the whole Chinese territory to produce 6 mean extreme rainfall maps (representative of 60 years with a time step of 10).

8. (L.398-402) Sometimes it is easy to blame the climate change for some catastrophic disasters. However, it was not seen any analysis in this study for climatic changes, even extreme rainfalls. Therefore, I think that it would be valuable to investigate the rainfall characteristics in each spatial and time clusters. Therefore, I again suggest that, daily rainfall characteristics such as 95 percentiles, 5-day max rainfall can be investigated in each cluster. However, if there is no data to investigate this phenomena, I would suggest open access datasets.

Indeed, the spatiotemporal distribution of extremes is know to be much more affected by climate change than its mean counterpart. Same as before, thank you for your suggestion. We have now fully implemented your comment in the new manuscript and computed the extremes rather than mean precipitation.

**Reviewer 2**

**General comments:**

This work focuses on spatiotemporal cluster analysis of flash flood disasters in China with the overall goal, as claimed by the authors, to identify and characterize spatiotemporal dependencies of flash flood related disasters. The scientific objectives of the manuscript are highly relevant to the scope of this journal and of interest to its readership. However, in my opinion, the current version of the manuscript does not succeed

to deliver scientific findings in a clear and convincing way. I provide below my major and minor comments that will hopefully help the authors to improve their work.

Thanks for your time and efforts to read and revise our paper. We do appreciate your positive comments and will take fully consideration on your suggestions.

**Major comment:**

1. How can we be sure that part of the findings is not related to the way the database was constructed? For example, can the duration of clusters shown as boxplots in Fig.9 be a result of infrequent records before 1970?

Indeed the reviewer is correct! Theoretically, part of the variability captured by our spatiotemporal clustering procedure can be explained through the way the dataset itself was built. This may be primarily due to the fact that data acquisition and report in the database was not fully operational in the earliest period our our temporal domain. However, here there is little that anyone can do. The only solution one has is to share this information with the readership as clear and transparent as possible. We actually did this, in our original manuscript (see L.310 314) exactly for the purpose of indicating a potential bias in the procedure.
Following your suggestion, we have extended our comments about the influence of the dataset itself on the results. This was added in the Discussions where we aimed at sharing our perspectives with the NHESS readership (L.380-384).

2. More effort needs to be done to link findings on the clusters with the physical meaning of flash flood related properties or occurrence. (P19, L385) Authors state "The most significant cluster resulting from the yearly model was detected in 1975", so what does this mean exactly? You need to improve a lot the interpretation of results in the current manuscript and help the reader understand what the different findings actually mean. Otherwise, the work will remain predominantly a cluster analysis with little information on the characteristics of the actual hazard.

A genuine thanks for this comments and suggestion. We have gone through the manuscript once more and we have done our best to improve the interpretative sections. For instance, with regards to the specific interpretation of the most significant cluster, we better explained this concept in the Methods (L.217-223), as follows:
"The cylinder with the highest GLR-value is the most likely cluster, that is, the cluster least likely to be due to chance, while the following are secondary clusters. Then, Monte Carlo simulations are performed and the statistical significance of the detected clusters can be assigned by comparing the rank of GLR from the real dataset with the GLR from the simulated one."
As well in the RESULTS, were we changed the sentence (L.281-290) as follows:
"To confirm this finding, we computed the temporal duration of the first ten clusters of flash flood disasters detected by applying a $T_{max}$ equals to 3 years and for the three models, defined by using values of $R_{max}$ equal to 100, 200 and 300 $km$ (Table3). Results confirm that clusters duration, expresses as start and end date, never exceed one year. The most significant cluster (ranked as ID=1) is the same for any model and dated to 1975. Secondary clusters (just from the second to the tenth) are exactly the same using $R_{max}$ of 200 or 300 $km$ while, reducing the radius at $100km$, their ranking can change, as for their size, since small clusters can merge into bigger ones by increasing $R_{max}$. Finally, it is worth noting that the top-ten clusters are well distributed over the entire study period, with the oldest one detected between 1963 and 1969 and the latest in 2010."

3. As a follow up of previous point, you present a number of findings (in terms of cluster attributes etc.) that add more confusion than clarity. Cluster characteristics for different temporal thresholds of 1,3,5 yrs or monthly models etc. are provided but interpretation/significance of each of those is not clearly delivered.

In light of this comment we have gone through the manuscript once more specifically aiming at clarifying the messages we shared with the readers. As a result, we have re-organised the section Results in order to first introduce some concepts and then to discuss only the most important findings related to those. And, we also looked into simplifying the text. For instance, the result of the statistics analysis (Section 3.2, Spatiotemporal clusters) have been re-structured into three new subsection namely:

- Subsection 3.2.1, Cluster characterization and their spatial distribution;
- Subsection 3.2.2, Temporal characterization of detected clusters;
- Subsection 3.2.3, Clusters pattern evolution at decade-scale.

4. (P16, L345) "these newly detected clusters can be due to the intensification of the extreme rainfall...". This can potentially be a very interesting point, but more work is needed to justify this. You would have to actually look at the precipitation record in the area and do an event-based analysis to find whether indeed extreme storm event are more frequent.

This comments is closely related to a comment already made from Rev1. We would like to thank you both because this specific comment has triggered stimulating discussions among us. As a result, we have now removed from the manuscript anything related to mean precipitation regimes. And, we have substituted this information with extreme rainfall. This has been computed per decade and shared with a new figure in the manuscript. Also, we have commented on this new data source and its behavior with respect to flash floods in the manuscript.

5. Definitions/explanation of certain terms and approaches is required. How is the term "repeated clusters" defined? How is "their relative occurrence (similarly to the concept of return period...)" determined?

We have now changed the term "*repeated clusters*" with "*overlapping clusters*". And, we have re-formulated the entire sentence to better explain this analysis, as follow: "Spatiotemporal clusters of flash flood disasters detected in China by decades were further assembled in a unique image. To this end, the centroid of each cluster (with reference to Fig. 8) was extracted and intersected with the catchment boundaries. Then, we computed the total number of clusters per catchment (Fig. 10a) as well the average interval of time at which two consecutive clusters have been detected in the

same catchment (Fig. 10b). It results that the catchments mainly affected by clusters of flash floods along several decades are mainly located in the southeast sector and essentially in the coastal mountains and that, on average, most of the cluster occur within an interval of 10-20 years."

6. How do you define "impact" in the context of events selected? I assume that impact is somehow related to exposure and therefore all the events analyzed are related to areas where there is exposure. So, this does not mean that in areas not shown on your map, there were no flash flood events occurring. I think this should be clarified to highlight once again that the analysis is highly related to the existing database.

We suppose that the reviewer is referring to L.282, or "Detected clusters where further analyzed by considering the impact of flash food disasters." We originally referred to the impact of flash flood disasters as the combination of fatalities and economic losses (see Table 1 in the original manuscript). However, in light of your comment as well as Rev1, we understood that this element was more of a distraction in the manuscript rather than a real source of scientific information to be shared with the NHESS readership.
As a result, we have finally decided to remove any information related to costs and fatalities from the manuscript. Because of this choice, we have also modified Fig.5 and Fig.8, where clusters were originally represented with a size that is proportional to the impact. More specifically, now there is no differentiation in symbols.

7. Conclusions need to be improved to provide more quantitative information on the overall findings. For example, elaborate more on what you found on the repeated occurrence per catchment.

Thank you for the comments and suggestions, we have done our best to improve the quantitative description of the results we obtained and shared our views on what we think about the repeated occurrences of flash floods per catchments, through time.

**Minor comments:**

8. For Figure 1 and other similar figures: results for the islands are not visible at all. Also, what does the dash line represents? My understanding is that certain island territories are under dispute among several countries so I would suggest caution on how you represent those.

Thank you for the comments. We would like to clarify that our study area is concentrated in the mainland China. The records on the islands (except Hainan Island) are not available temporally. Therefore, we could not perform any analyses over these areas. As for what the dashed line represents, this is the Chinese geo-political boundary. It is true that the territory boundary is of vital important and we will absolutely be caution on it. We have also provided more details in the captions.

9. Fig.4: a legend to explain colors is missing.

Thank you very much for pointing this out. In Fig.4, each color indicates one unique cluster. We have now added this information to the caption to avoid any misunderstanding.